# A Multi-Scale Inception-UNet with Structure-Aware Evaluation for Branch-Preserving Segmentation of Organoids

**Author Sandra H. Andrusca**[1] (iD)                    SANDRA.ANDRUSCA@TUM.DE
**Author Christopher D. Kießling** [1]                   CHRISTOPHER.KIESSLING@TUM.DE
**Author Andreas R. Bausch** [1]                         ABAUSCH@MYTUM.DE

[1] *Heinz Nixdorf Chair in Biophysical Engineering of Living Matter*
*Center for Protein Assemblies (CPA)*
*Technical University of Munich (TUM)*
*85748 Garching, Germany*

**Editors:** Under Review for MIDL 2026

## Abstract

Branched organoids exhibit increasingly complex morphologies as they progress from simple spheroid states to highly ramified structures, making topology-preserving segmentation essential for quantitative biological analysis. Capturing thin protrusions and maintaining branch continuity remains challenging for classical UNet-based architectures, particularly in brightfield imaging where fine structures are easily blurred or disconnected.

In this work, we present a multi-scale Inception-UNet designed to capture the heterogeneous spatial scales of branched organoids through parallel convolutional paths with complementary receptive fields. As a model system, we analyze brightfield pancreatic ductal adenocarcinoma (PDAC) organoids, a system known for strong morphological heterogeneity and invasive branching behavior (Randriamanantsoa, 2022), cultured using high-throughput Patternoid assays (Kurzbach, 2025) that enable standardized imaging and robust quantitative analysis.

To assess segmentation quality beyond region overlap, we combine Dice with the structure-aware clDice metric that directly probes branch integrity and topological continuity. Across deterministic seeds and strictly separated organoid positions, the Inception-UNet achieves the highest region-based Dice ($0.868 \pm 0.062$) and clDice ($0.545 \pm 0.123$), and most importantly, the strongest preservation of branch continuity compared to UNet and UNet++. These improvements become increasingly pronounced with growing morphological complexity.

Overall, our results demonstrate that multi-scale feature extraction combined with topology-aware evaluation substantially improves segmentation of branched organoids and provides a robust foundation for downstream morphological and invasion-related analyses.

**Keywords:** semantic segmentation, organoid imaging, PDAC, multi-scale architectures, UNet, topology preservation, topology-aware loss, clDice, brightfield microscopy, multi-seed evaluation, branch morphology

## 1. Introduction

Branched organoids have emerged as an important model system for studying epithelial morphogenesis, invasion, and structural organization. As these systems transition from compact spheroids into highly ramified architectures, branch integrity and topological structure become central descriptors of their biological behavior and downstream quantitative

analyses. (Clevers, 2016; Boj, 2015). Pancreatic ductal adenocarcinoma (PDAC) organoids in particular exhibit pronounced morphological heterogeneity and complex, invasive branching dynamics (Randriamanantsoa, 2022), making them a demanding but informative benchmark for topology-preserving segmentation.

In this work, we analyze PDAC organoids imaged using high-throughput Patternoid assays (Kurzbach, 2025), which provide standardized 3D culture conditions and longitudinal brightfield acquisition across large organoid populations. The resulting morphologies contain both thin protrusions and broader structural compartments, requiring segmentation methods that can jointly capture fine-scale and coarse-scale features.

Classical UNet architectures (Ronneberger et al., 2015) remain the dominant approach for biomedical image segmentation, yet their single-scale convolutional blocks often struggle to preserve thin branches in brightfield organoids. UNet++ (Zhou et al., 2018) improves feature propagation through nested skip connections, while Inception-style multi-scale representations (Szegedy et al., 2015) and residual designs (He et al., 2016) have proven effective for capturing heterogeneous spatial patterns in natural images. However, these ideas have not been systematically adapted to branched organoid morphologies, where preserving topological continuity is essential for downstream quantitative analysis.

To address these limitations, we introduce a multi-scale Inception-UNet that integrates parallel convolutional paths with complementary receptive fields within each encoder stage. This design allows the network to model both thin, elongated branches and larger morphological compartments simultaneously. Because region-based Dice (Long et al., 2015) primarily reflects area overlap and is insensitive to many topological defects, we complement it with the structure-aware clDice metric that directly probes centerline continuity and branch preservation. All models are trained within a deterministic multi-seed PyTorch pipeline (Paszke et al., 2019) to ensure reproducibility.

Under identical training and evaluation settings with strictly separated organoid positions, the Inception-UNet achieves the highest Dice and skeleton-based Dice among all compared architectures and shows the clearest improvements in branch continuity. These results highlight the importance of multi-scale feature extraction and topology-aware evaluation for analyzing the complex morphologies of branched organoids.

Our contributions are threefold:

- We introduce a multi-scale Inception-UNet architecture tailored to the branched morphology of PDAC organoids cultured in high-throughput Patternoid assays, and evaluate it within a deterministic multi-seed analysis pipeline with strictly separated organoid positions.

- We establish a topology-aware evaluation protocol that combines region-based Dice with a structure-aware clDice metric to specifically probe branch continuity and structural integrity beyond conventional overlap measures.

- We provide an initial analysis of a skeleton-aware auxiliary loss that augments standard region-based supervision with topology-focused signals, illustrating how explicit branch-aware objectives can further enhance structural fidelity.

## 2. Materials and Methods

### 2.1. Dataset

We analyze brightfield microscopy data of pancreatic ductal adenocarcinoma (PDAC) organoids, characterized by heterogeneous and invasive growth patterns (Randriamanantsoa, 2022). The dataset is generated using the Patternoid high-throughput platform (Kurzbach, 2025), enabling standardized culture and imaging.

Organoids are recorded as hourly time-lapse sequences, each consisting of a brightfield $z$-stack (~100 slices). For segmentation, stacks are converted into 2D standard-deviation (STD) projections computed over in-focus slices, enhancing local contrast and improving visibility of thin branches.

The dataset spans three PDAC morphological subtypes (Boj, 2015; Clevers, 2016), ranging from compact to highly branched structures. Since we address binary segmentation, all subtypes are pooled to increase morphological diversity.

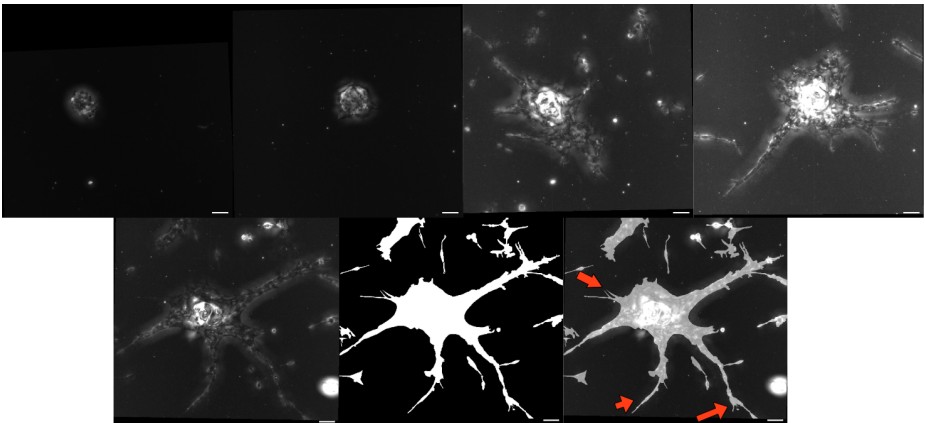

Figure 1: Representative STD projections of PDAC organoids across four time windows, illustrating the transition from compact to highly branched morphologies. Bottom: example with ground truth and overlay highlighting thin branches. Scale bar: 50 μm.

To avoid temporal and positional leakage, we enforce strict separation of organoid positions: all time points from a given organoid are assigned to exactly one of the training, validation, or test sets. Evaluation is therefore performed on entirely unseen organoids rather than additional time points of already observed ones.

Manual pixel-wise annotations were generated by trained experts on the projected images. The final dataset comprises 361 training images, 101 validation images, and 81 test images, corresponding to distinct organoid instances under the position-based split described above.

## 2.2. Preprocessing

Each $z$-stack is rigidly registered to correct for drift, followed by selection of in-focus slices and computation of a 2D STD projection. This projection enhances structural details, particularly thin protrusions.

To preserve spatial resolution, images are not resized. Instead, each frame is padded to $768 \times 768$ pixels and processed using overlapping $256 \times 256$ tiles. During training, tiles are sampled using a foreground-aware strategy, while inference uses weighted stitching to reconstruct full-resolution predictions without boundary artifacts.

## 2.3. Inception-UNet Architecture

The proposed Inception-UNet extends the standard UNet by replacing encoder blocks with lightweight multi-scale Inception modules (Figure 2). These modules combine parallel convolutions with different receptive fields to capture both local and global context.

Each encoder block consists of two branches: a $3 \times 3$ convolution for local features and a larger receptive-field branch approximating $7 \times 7$ context. The outputs are concatenated and fused via a $1 \times 1$ convolution, followed by normalization and non-linearity.

To enhance feature representation, we incorporate channel-wise recalibration (SE blocks) and residual connections, improving stability and preserving fine spatial details.

The decoder follows the standard UNet design with upsampling and skip connections. All models are matched in depth and parameter scale to ensure a fair comparison.

## 2.4. Structure-Aware Evaluation

Region-based metrics such as Dice primarily quantify area overlap but are insensitive to topological errors, particularly in thin and elongated structures. For branched PDAC organoids, such errors can affect downstream analyses of branch continuity and morphology.

To capture structural fidelity, we complement Dice with the clDice metric, which evaluates overlap of soft skeleton representations and is sensitive to discontinuities and missing branches.

In addition, we consider a connectivity-based penalty that quantifies local continuity along predicted skeletons. This term is used to derive an aggregated topology score:

$$\text{TopoScore} = \text{clDice} - \lambda \cdot \text{GraphPenalty}. \tag{1}$$

This score provides a complementary perspective by combining overlap-based and connectivity-based aspects of structural quality.

Together, these metrics enable a more comprehensive evaluation, revealing structural differences that are not captured by region-based Dice alone.

## 2.5. Exploratory Topology-Aware Loss

We explore whether explicit topology-aware supervision can improve structural fidelity during training.

The baseline loss combines Dice and binary cross-entropy:

$$\mathcal{L}_{\text{reg}} = \alpha \, \mathcal{L}_{\text{Dice}} + \beta \, \mathcal{L}_{\text{BCE}}. \tag{2}$$

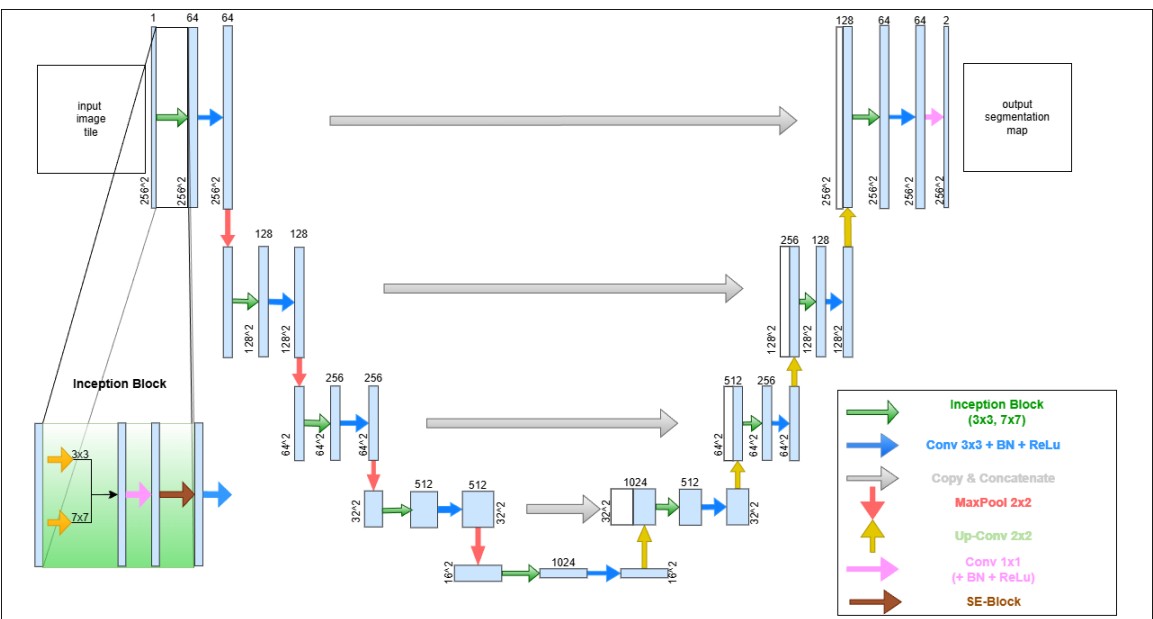

Figure 2: Overview of the proposed Inception-UNet architecture. Each encoder stage replaces standard convolutional blocks with a two-branch multi-scale module consisting of a local $(3 \times 3)$ convolution and a larger receptive-field branch approximating a $(7 \times 7)$ context. The branch outputs are combined and projected via a $(1 \times 1)$ convolution, followed by batch normalization and ReLU activation. Channel-wise feature responses are recalibrated using a squeeze-and-excitation (SE) block, and residual connections are employed to stabilize training and preserve fine spatial details. The decoder follows the standard UNet design with upsampling, skip connections, and $(3 \times 3)$ refinement convolutions.

To incorporate structural information, we add auxiliary terms operating on soft skeleton representations. A soft skeleton Dice term encourages alignment of predicted and reference centerlines:

$$\mathcal{L}_{\text{skel}} = 1 - \text{Dice}(S_{\text{pred}}, S_{\text{gt}}), \tag{3}$$

and a continuity term penalizes fragmented predictions:

$$\mathcal{L}_{\text{graph}} = \frac{1}{|\Omega|} \sum_{p \in \Omega} \max(0, \tau - N(p)). \tag{4}$$

The final loss is defined as

$$\mathcal{L} = \alpha \, \mathcal{L}_{\text{Dice}} + \beta \, \mathcal{L}_{\text{BCE}} + \gamma \, \mathcal{L}_{\text{skel}} + \delta \, \mathcal{L}_{\text{graph}}. \tag{5}$$

These terms are evaluated as exploratory extensions. While they can improve structure-aware metrics, they do not consistently outperform simpler formulations and may lead to slightly inflated segmentations. We therefore retain the simpler loss for the main experiments.

## 2.6. Training Procedure

All models are trained under a deterministic multi-seed protocol using seeds $\{0, 1, 2, 3, 4\}$. Random number generators in Python, NumPy, and PyTorch are initialized per seed to control stochastic effects, while allowing non-deterministic backend optimizations for computational efficiency.

Models are optimized using Adam with fixed hyperparameters shared across all architectures. Hyperparameters are selected once based on a reference UNet model and kept constant to ensure a fair comparison.

Training is performed with early stopping based on validation performance. For ablation studies, models are trained for up to 100 epochs (patience 15), while final experiments use up to 150 epochs (patience 30) to ensure convergence.

Model selection is based on the best validation score.

## 2.7. Baseline Models

We compare the proposed Inception-UNet with two widely used architectures for biomedical image segmentation: UNet (Ronneberger et al., 2015) and UNet++ (Zhou et al., 2018).

To ensure a fair comparison, all models are matched in encoder–decoder depth, channel width, and overall parameter scale.

**UNet.** UNet follows a standard encoder–decoder architecture with symmetric skip connections for multi-scale feature fusion.

**UNet++.** UNet++ extends UNet with nested skip connections that refine encoder features before fusion in the decoder.

**Training Conditions.** All models are trained under identical settings, including optimizer, learning rate, batch size, data splits, and deterministic multi-seed protocol. Unless stated otherwise, models are optimized using a Dice + BCE loss, optionally augmented with topology-aware terms (e.g., clDice).

## 3. Results

### 3.1. Quantitative Evaluation

We evaluate UNet, UNet++, and the proposed Inception-UNet on the held-out test set using region-based Dice and the structure-aware clDice metric (Section 2.4). All values are reported as mean ± standard deviation across deterministic seeds.

Statistical significance is assessed using paired Wilcoxon signed-rank tests on per-image values averaged across seeds (mean-over-seeds).

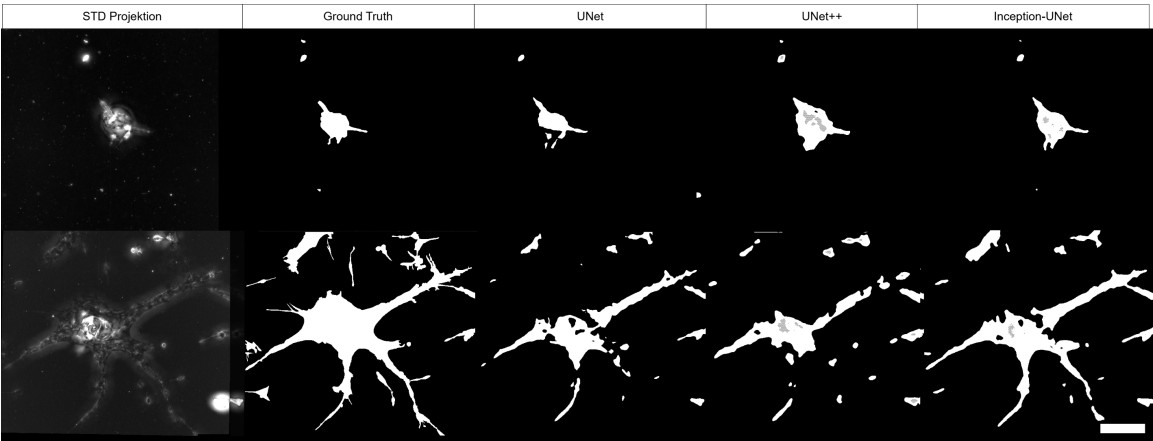

Figure 3: Qualitative comparison of segmentation performance on PDAC organoids. From left to right: STD projection, ground truth, UNet, UNet++, and Inception-UNet. Top row: example of a small organoid. UNet under-segments the structure and fails to capture its full extent, while UNet++ tends to over-segment and produces inflated masks. The Inception-UNet more accurately recovers the overall shape and boundary. Bottom row: highly branched organoid. Both UNet and UNet++ produce fragmented predictions and fail to preserve thin branches, particularly along the lower protrusion. In contrast, the Inception-UNet yields more continuous segmentations and better maintains branch connectivity. Scale bar: 100 μm.

The Inception-UNet achieves the highest Dice scores across seeds. Wilcoxon testing confirms significant improvements over both UNet and UNet++ ($p < 0.001$, $r > 0.8$).

For structure-aware evaluation, skeleton-based Dice reveals clearer separations between models. The Inception-UNet significantly outperforms both baselines ($p < 0.01$), indicating improved preservation of thin and elongated structures.

Overall, the proposed model improves both region overlap and structural fidelity compared to standard architectures.

Table 1 summarizes the overall segmentation performance across models, showing that the Inception-UNet achieves the highest scores across all metrics, with the largest improvements observed in structure-aware measures.

Table 1: Segmentation performance (mean ± std over seeds).

| Model | Dice | clDice | TopoScore |
|---|---|---|---|
| UNet | 0.862 ± 0.076 | 0.438 ± 0.168 | 0.309 ± 0.214 |
| UNet++ | 0.828 ± 0.094 | 0.442 ± 0.163 | 0.043 ± 0.208 |
| Inception-UNet | **0.868 ± 0.062** | **0.545 ± 0.123** | **0.311 ± 0.193** |

## 3.2. Structural Fidelity Beyond Region Overlap

Region-based Dice captures overlap but fails to reflect topological errors such as broken or disconnected branches. As shown in Figure 3, UNet and UNet++ frequently produce discontinuities in thin protrusions, whereas the Inception-UNet maintains more consistent centerlines and connectivity.

These differences are captured by skeleton-based Dice, which reveals structural discrepancies that remain largely invisible to region-based metrics.

## 3.3. Stability Across Seeds

All models show low seed-to-seed variation, indicating stable training. The relative ranking remains consistent, with Inception-UNet achieving the best performance across metrics.

## 3.4. Ablation Studies

We perform controlled ablations to assess the impact of preprocessing, loss design, and architecture on segmentation performance.

### 3.4.1. RESOLUTION ABLATION

We compare a conventional downsampling approach ($128 \times 128$) with a resolution-preserving pad-and-tile strategy ($768 \times 768$ with $256 \times 256$ patches).

The resolution-preserving approach consistently improves performance, particularly for structure-aware metrics. While region-based Dice shows only moderate differences, topology-aware scores benefit substantially from maintaining native spatial resolution.

These results highlight that aggressive downsampling degrades the representation of thin and elongated structures, leading to reduced topological fidelity.

### 3.4.2. LOSS ABLATION

To isolate the effect of topology-aware supervision, we perform a loss ablation study using a fixed UNet architecture under identical training conditions.

We compare a region-based baseline (BCE + Dice) with topology-aware extensions, including clDice and a more complex custom formulation combining skeleton and graph constraints.

Region-based Dice remains largely unchanged across all configurations, whereas structure-aware metrics show clear differences. The addition of clDice consistently improves structural fidelity, while more complex loss formulations do not provide further benefits and may introduce slightly thicker segmentations.

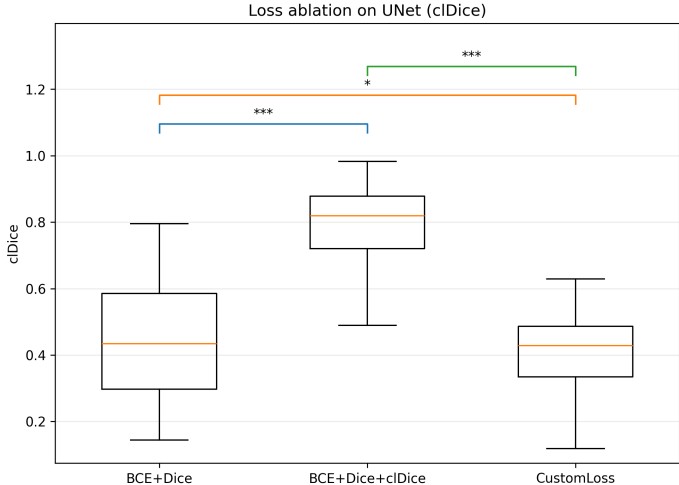

Figure 4: Loss ablation on a fixed UNet architecture using the structure-aware clDice metric. Boxplots show per-image clDice scores on the test set, averaged across five deterministic seeds (mean-over-seeds). Statistical significance is assessed using paired Wilcoxon signed-rank tests. The addition of clDice significantly improves structural performance over the region-based baseline ($p < 0.001$), while the more complex custom loss does not yield further gains.

Based on these results, we retain the simpler **BCE + Dice + clDice** formulation for subsequent experiments.

### 3.4.3. Architecture Ablation

To isolate the effect of network architecture, we compare UNet, UNet++, and the proposed Inception-UNet under identical training conditions, including preprocessing, data splits, and loss formulation.

All models are matched in encoder–decoder depth and parameter scale to ensure a controlled comparison.

Across architectures, region-based Dice remains comparable, whereas structure-aware metrics reveal clear differences. The Inception-UNet consistently achieves higher clDice scores, indicating improved preservation of thin branches and connectivity.

Importantly, these gains are not caused by inflated segmentations, but reflect more coherent centerline reconstruction, as confirmed by qualitative inspection.

Based on these results, we select the Inception-UNet for subsequent experiments.

### 3.5. Final Model Comparison and Statistical Analysis

We compare UNet, UNet++, and the proposed Inception-UNet using paired Wilcoxon signed-rank tests on the test set. Per-image metrics are first averaged across deterministic training seeds (mean-over-seeds), yielding one value per image and model.

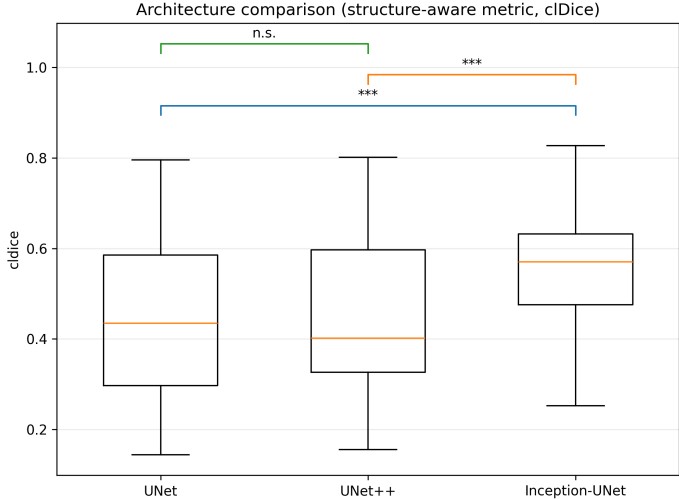

Figure 5: Architecture comparison using the structure-aware clDice metric. Boxplots show per-image clDice scores on the test set, averaged across five deterministic seeds (mean-over-seeds). Statistical significance is assessed using paired Wilcoxon signed-rank tests. The proposed Inception-UNet significantly outperforms both UNet and UNet++ ($p < 0.001$), while no significant difference is observed between UNet and UNet++ (n.s.).

The Inception-UNet achieves significantly higher region-based Dice scores than both UNet and UNet++ ($p < 10^{-10}$, $r > 0.85$), indicating consistent improvements across the dataset.

For structure-aware metrics, Inception-UNet also significantly outperforms both baselines ($p < 0.01$), confirming improved preservation of thin branches and connectivity. While absolute differences are smaller, effect sizes remain moderate to large, indicating systematic improvements rather than isolated cases.

Importantly, these gains are not accompanied by inflated segmentations. Qualitative inspection shows that Inception-UNet preserves coherent centerlines while maintaining compact foreground regions.

## 3.6. Exploratory Topology-Aware Loss

We additionally evaluate a skeleton-aware loss that combines region-based supervision with explicit skeleton and connectivity constraints. While this formulation improves clDice in some cases, it does not consistently outperform the simpler clDice-based loss and can lead to slightly inflated segmentations.

We therefore retain the BCE + Dice + clDice formulation for the main experiments.

## 4. Discussion

We demonstrate that topology-preserving segmentation of highly branched PDAC organoids from brightfield microscopy is feasible despite low contrast and limited annotations. While region-based Dice remains similar across models, structure-aware metrics reveal substantial differences in the preservation of thin branches and connectivity.

The proposed Inception-UNet achieves the best overall performance, particularly in later, highly branched stages. This suggests that multi-scale feature extraction is beneficial for capturing heterogeneous spatial structures. In contrast, UNet++ does not provide consistent advantages in this setting, indicating that nested skip refinements are less effective under limited structural contrast.

Our results further highlight the importance of topology-aware evaluation. The clDice metric reveals structural differences that are not captured by region overlap alone, emphasizing the need for complementary evaluation strategies.

Overall, we establish a reproducible benchmark for topology-aware segmentation of brightfield organoids and show that multi-scale architectures improve structural fidelity without sacrificing compactness.

## 5. Outlook

Future work will explore more advanced topology-aware losses and evaluation metrics to better capture subtle structural defects in complex organoid morphologies.

## Acknowledgments

We thank Sophie Kurzbach for providing the experimental data. We acknowledge the support by the Deutsche Forschungsgemeinschaft (DFG, German Research Foundation) through the project BA 2029/15-1 and by the European Research Council under the European Union's Horizon 2020 research and innovation programme (grant agreement no. 810104-PoInt).

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

## Appendix A. Topology-Aware Loss Formulations

In addition to standard region-based supervision, we explored topology-aware loss formulations to encourage structural consistency in thin and elongated organoid branches. This section summarizes the loss components used in the loss ablation study (Section 3.4.2).

### A.1. Region-Based Loss

The baseline loss combines binary cross-entropy (BCE) and soft Dice loss:

$$\mathcal{L}_{\text{reg}} = \alpha \, \mathcal{L}_{\text{Dice}} + \beta \, \mathcal{L}_{\text{BCE}}. \tag{6}$$

### A.2. Soft Skeleton Overlap (clDice)

To incorporate topology-aware supervision without requiring hard skeletonization, we employ the clDice formulation:

$$\mathcal{L}_{\text{clDice}} = 1 - \frac{2 \, T_{\text{prec}} \cdot T_{\text{sens}}}{T_{\text{prec}} + T_{\text{sens}} + \varepsilon}, \tag{7}$$

with

$$T_{\text{prec}} = \frac{|S_{\text{pred}} \cap M_{\text{gt}}|}{|S_{\text{pred}}| + \varepsilon}, \tag{8}$$

$$T_{\text{sens}} = \frac{|S_{\text{gt}} \cap M_{\text{pred}}|}{|S_{\text{gt}}| + \varepsilon}. \tag{9}$$

### A.3. Custom Topology Loss (Skeleton + Graph)

As an alternative, we define a custom topology-aware loss that combines explicit skeleton supervision with a connectivity constraint:

$$\mathcal{L}_{\text{topo}} = \mathcal{L}_{\text{skel}} + \mathcal{L}_{\text{graph}}. \tag{10}$$

The skeleton supervision term is given by

$$\mathcal{L}_{\text{skel}} = \text{BCE} \left( S_{\text{pred}}^{\text{hard}}, S_{\text{gt}}^{\text{hard}} \right), \tag{11}$$

where $S^{\text{hard}}$ denotes binary skeletons obtained via morphological thinning.

The graph-based continuity term penalizes insufficient local connectivity:

$$\mathcal{L}_{\text{graph}} = \frac{1}{|\Omega|} \sum_{p \in \Omega} \max \left( 0, \tau - N(p) \right), \tag{12}$$

where $N(p)$ is the number of active neighbors of pixel $p$, and $\tau$ is a minimum connectivity threshold.

## A.4. Final Training Objective

The overall loss is defined as

$$\mathcal{L} = \alpha\,\mathcal{L}_{\text{Dice}} + \beta\,\mathcal{L}_{\text{BCE}} + \gamma\,\mathcal{L}_{\text{topo}}, \tag{13}$$

where $\mathcal{L}_{\text{topo}}$ corresponds either to $\mathcal{L}_{\text{clDice}}$ or the custom topology loss defined above.

All topology-aware losses are treated as exploratory extensions and are evaluated in the ablation study in Section 3.4.2.

