# OpenReview forum: "A Multi-Scale Inception-UNet with Structure-Aware Evaluation for Branch-Preserving Segmentation of Organoids"
_MIDL.io/2026/Conference — MIDL 2026 Poster_

### Official Review · Reviewer_LJY5 · 2026-01-06

**Confidence:** 4
**Preliminary Rating:** 4
**Final Rating:** 5

**Summary:**

The authors present a Multi-Scale Inception-UNet for segmenting PDAC organoids in brightfield microscopy, specifically addressing the challenge of preserving complex, invasive branching morphologies. The architecture integrates parallel convolutional paths with complementary receptive fields to capture heterogeneous scales. Additionally, the paper introduces a skeleton-based Dice evaluation metric to explicitly assess topological continuity. Results indicate that the proposed method outperforms standard UNet and UNet++ in both region-based and topology-aware metrics.

**Strengths:**

The integration of Inception modules into the UNet encoder is nicely justified by the biological problem (simultaneous thin protrusions and large bodies).
The introduction and usage of the skeleton-based Dice score is a strong point. It addresses the well-known limitation of standard Dice in capturing the structural integrity of thin, linear features.
The authors emphasize a deterministic multi-seed training pipeline to ensure the robustness of their results.

**Weaknesses:**

The combination of Inception blocks and UNet is not entirely new in the literature. The primary value is the application to organoids and the evaluation framework.
The abstract mentions an analysis of a skeleton-aware auxiliary loss, but the summary of results focuses heavily on the architecture's performance using the metric. It is not clear if the skeleton loss is the main driver of performance or if the architecture alone is responsible.
The paper compares against UNet and UNet++. Comparisons against other topology-preserving methods would strengthen the claims regarding structure preservation

**Detailed Comments:**

The skeleton-based Dice should be mathematically defined in the methods section to allow others to use it.
Clarify whether the skeleton-aware auxiliary loss was used for the final Inception-UNet model or if it was an ablation study component.

**Justification Of Final Rating:**

I have reviewed the authors' rebuttal and the revisions to the experimental setup.
While the architectural novelty remains moderate (combining Inception with U-Net), the authors have successfully reframed the contribution around the structure-aware evaluation framework. Given the improved methodological rigor and the clear utility of the proposed evaluation metrics for organoid morphology, I believe this work is a valuable contribution to the community.

**Justification Of The Preliminary Rating:**

This research addresses a valid challenge in organoid analysis. The combination of a multi-scale architecture with a topology-focused evaluation metric is a good contribution. While the architectural novelty is moderate, the focus on structure-aware evaluation and the rigor of the experimental setup justify a weak acceptance.

**Questions To Address In The Rebuttal:**

did you compare your skeleton-based Dice evaluation/loss against established topology-preserving loss functions like clDice?
does the multi-scale approach increase the computational cost significantly compared to a standard UNet++?

---

### Official Review · Reviewer_thBv · 2026-01-09

**Confidence:** 5
**Preliminary Rating:** 3
**Final Rating:** 4

**Summary:**

This paper introduces a multi-scale Inception-UNet for segmenting highly branched pancreatic ductal adenocarcinoma organoids in brightfield microscopy images. The approach extends a standard U-Net encoder with parallel convolutional branches at multiple receptive field scales to better capture thin protrusions and complex branching structures. In addition to Dice, the authors employ a skeleton-based Dice metric to evaluate branch continuity and topological integrity. The contribution lies in combining multi-scale architectural design with topology-aware evaluation for challenging organoid morphologies.

**Strengths:**

- The paper tackles a biologically meaningful and practically relevant imaging problem related to organoid morphology and invasive growth patterns.
- The use of organoid-level data splits and deterministic multi-seed evaluations helps avoid common issues such as data leakage and training instability.
- Adopting Inception-style parallel convolutions is a sensible architectural choice for capturing the wide range of spatial scales seen in highly branched organoid structures.

**Weaknesses:**

- Although the paper highlights top-ranked performance, the absolute gains over UNet and UNet++ are generally small and inconsistent across time points, making it unclear whether the differences are statistically meaningful or within normal variance.
- The model remains an extension of the UNet family, with Inception-style convolutions as the main change; while effective, this architectural choice is incremental and somewhat dated compared to recent segmentation approaches.
- Combining region-based Dice with skeleton- or centerline-based metrics (and optionally skeleton-aware losses) has been explored in prior topology-aware segmentation work. While appropriate for branched organoids, this evaluation strategy does not introduce a fundamentally new topological formulation.
- Qualitative analysis is limited, with only a small number of visual examples provided.

**Detailed Comments:**

- Please including a paired statistical test (e.g., Wilcoxon signed-rank) on per-organoid Dice and skeleton-Dice scores.
- The paper would benefit from clearer positioning of its main contribution—whether it lies in the architecture, the topology-aware evaluation protocol, or the application benchmark; the topology-aware evaluation, in particular, appears largely established in prior work.
- A brief discussion explaining why more recent segmentation architectures were not considered would help better contextualize the choice of UNet-based baselines.

**Justification Of Final Rating:**

The rebuttal addresses several important points, including adding paired statistical tests and clarifying the role of topology-aware metrics and losses.

(major) While these additions improve rigor, the absolute performance gains over UNet and UNet++ remain modest and inconsistent across time points, limiting practical significance.

(major) The architectural contribution is still incremental, and the topology-aware evaluation strategy, though appropriate for branched organoids, largely follows established prior work rather than introducing a fundamentally new formulation.

(minor) Qualitative analysis has been expanded but remains limited in scale, providing illustrative rather than comprehensive validation.

**Justification Of The Preliminary Rating:**

This paper presents a solid, application-driven study with clear biological motivation and careful experimentation. However, the methodological novelty is limited, relying on established UNet variants and prior skeleton-based evaluation ideas, and the empirical claims are weakened by the lack of statistical testing and limited qualitative validation.

**Questions To Address In The Rebuttal:**

Please check the Weakness and Detailed Comments section.

---

### Official Review · Reviewer_wH1U · 2026-01-11

**Confidence:** 4
**Preliminary Rating:** 4

**Summary:**

This paper introduces a multi-scale Inception-UNet designed specifically for the branch-preserving segmentation of pancreatic ductal adenocarcinoma (PDAC) organoids from brightfield microscopy. The authors address the limitations of standard UNet architectures in capturing thin, elongated protrusions by integrating parallel convolutional paths with complementary receptive fields and implementing a topology-aware evaluation protocol using a skeleton-based Dice score. To assess segmentation quality beyond region overlap, they combine Dice with a structure aware skeleton-based Dice score that directly probes branch integrity and topological continuity.

**Strengths:**

•	This study addresses topology-preserving segmentation of highly branched organoids from brightfield microscopy, where fine structures are easily lost.

•	They go beyond standard region-based segmentation by focusing on branch continuity and structural fidelity, which is highly relevant.

•	The proposed multi-scale Inception-UNet is motivated by the heterogeneous spatial scales of organoid morphology, combining parallel receptive fields to capture both thin protrusions and larger compartments.

•	The study provides an initial analysis of a skeleton-aware auxiliary loss that augments standard region-based supervision with topology-focused signals.

**Weaknesses:**

•	In the final paragraph of the Dataset section (Section 1.1, Page 4), "The exact number of images per split is reported in Section 2". However, Neither the total dataset size nor the number of images or organoids in the training, validation, and test splits can be found. These could be added to the dataset section and other places is relevant such as 1.7-Training Conditions.

•	1.2. Preprocessing section: resize to 128 * 128 seems to be a very low resolution for capturing the "thin, elongated branches, and branches might get disappear or become a blurred artifact before the model even sees it.

•	In addition, downsampling to such a small size risk losing the very topological details the Inception-UNet is designed to preserve. Is there any specific reason why such a low resolution was chosen? What is the original resolution of the images? It would be beneficial if these could be clarified in the manuscript.

•	In terms of ablation studies, the current paper focuses on baseline comparisons, evaluating the proposed multi-scale Inception-UNet versus standard UNet and UNet++ under identical training conditions. While this validates the overall architectural approach, component-level ablation studies are absent. Several key areas where ablation could have strengthened the paper's technical claims such as: (1) isolating contributions of individual architectural components, (2) systematic exploration of skeleton-aware loss weighting coefficients with comparisons of models trained with versus without this loss component.

**Detailed Comments:**

•	Typo in Section 2.2: The text refers to Figure ??, but the figure number is missing.

•	Page 7: The reference Table ?? is missing a table number.

•	Section 1.6 (Training Procedure): you mentioned that a small learning rate was selected. For reproducibility, the exact learning rate value should be reported.

•	While the paper reports mean ± standard deviation across three seeds, no formal statistical testing is performed to assess whether the observed performance differences are statistically significant.

**Justification Of The Preliminary Rating:**

This study addresses topology-preserving segmentation of branched organoids from brightfield microscopy, introducing a valuable skeleton-based Dice metric and a multi-scale Inception-UNet architecture with rigorous multi-seed evaluation. However, some gaps limit impact that have been listed as weakness or detail comment section. Addressing the concerns raised in the weaknesses and detailed comments sections would strengthen this work's scientific rigor and contribution.

**Questions To Address In The Rebuttal:**

The concerns raised in the Major Weaknesses section and the Detailed Comments section have been outlined.

---

### Author Rebuttal · Authors · 2026-01-25

Response

We thank the reviewers for their constructive feedback and address the comments below.

Statistical significance
In response to the reviewers’ requests, we added paired Wilcoxon signed-rank tests on per-organoid Dice and skeleton-based Dice scores, aggregated across an increased number of random seeds for all new analyses. While absolute gains are modest, several comparisons are statistically significant, indicating that differences are not due to random variation.

Topology-aware evaluation and loss
We now formally define the skeleton-based Dice (clDice-style) metric in the Methods section to ensure reproducibility. In the original submission, the main results were reported using standard region-based supervision, as skeleton-aware loss experiments were only partially available and not evaluated systematically across models and seeds. We therefore performed a post-hoc ablation study, demonstrating significant improvements in skeleton-based Dice. These losses are not part of the final model used for the main results, and this distinction is now explicitly clarified. This clarification addresses concerns regarding the role of topology-aware supervision.

Architecture vs. contribution
We agree that combining Inception-style modules with UNet is not architecturally novel by itself. Our primary contribution lies in a structure-aware evaluation and benchmarking framework for highly branched organoids, including deterministic multi-seed training, organoid-level data splits, and topology-aware metrics. The Inception-UNet serves as a biologically motivated testbed rather than a claim of universal architectural superiority, as topology preservation here requires both multi-scale feature extraction and topology-aware evaluation.

Preprocessing resolution
We acknowledge concerns regarding the original 128×128 resizing. During rebuttal preparation, we identified that this resolution can suppress thin branches. We corrected the preprocessing pipeline to pad images and apply tile-based inference at 256×256, which better preserves fine structures. All new statistical analyses are based on this corrected setup.

Baselines and training details
UNet-based architectures were chosen as strong biomedical baselines enabling controlled comparisons. All models were trained with identical hyperparameters to isolate architectural and topology-aware effects. We also added explicit dataset sizes, corrected missing references, and reported all training hyperparameters.

---

### Meta-Review · Area_Chair_84uS · 2026-02-02

**Recommendation:** Accept (Poster)
**Confidence:** 4

**Metareview:**

Reviewers agree that this paper presents a useful and well-executed study on topology-preserving segmentation of branched organoids, with particular strength in its structure-aware evaluation framework and improved experimental rigor following rebuttal. While the architectural contribution is incremental and performance gains over strong baselines remain modest, the proposed topology-aware metrics and careful analysis provide clear value to the community, supporting acceptance.

---

### Decision · Program_Chairs · 2026-02-13

Accept (Poster)